# Two-years of stratospheric chemistry perturbations from the 2019/2020 Australian wildfire smoke

Kane Stone[1], Susan Solomon[1], Pengfei Yu[2], Daniel M. Murphy[3], Douglas Kinnison[4], Jian Guan[1]

[1]Department of Earth, Atmospheric, and Planetary Sciences, Massachusetts Institute of Technology, Cambridge, MA 02139, USA
[2]Institute for Environmental and Climate Research, College of Environment and Climate, Jinan University, Guangzhou, China
[3]Chemical Sciences Laboratory, National Oceanic and Atmospheric Administration, Boulder, CO 80305, USA
[4]Atmospheric Chemistry Observations & Modeling Laboratory, NSF National Center for Atmospheric Research, Boulder, CO 80307, USA

*Correspondence to*: Kane Stone (stonek@mit.edu)

## Abstract

The very large pyrocumulonimbus events that occurred during the Australian summer of 2019/2020 caused extremely unusual partitioning of stratospheric chlorine in Southern Hemisphere midlatitudes and Antarctic regions not only in 2020 but also in 2021. This was likely caused by enhanced HCl solubility in organic species that increased heterogeneous chemistry. Here, we show that observed HCl and $ClONO_2$ values remain outside the pre-wildfire satellite range since 2005 in both the Southern Hemisphere midlatitude and Antarctic regions in 2021. Through model simulations, we replicate this multi-year prolonged chemical perturbation, in good agreement with observations. This was achieved by calculating HCl solubility in mixed wildfire and sulfate aerosols consistent with assumptions of 1) liquid-liquid phase separation and 2) linear dependence on organic and sulfate composition. The model simulations also suggest that the Australian pyrocumulonimbus organic aerosols contributed to low midlatitude ozone values in 2020 and 2021. A marked photochemically controlled seasonality of the chemical perturbations and ozone depletion is also observed and simulated, and its underlying chemical drivers are identified. This work highlights that lower concentrations of smoke still had profound effects on stratospheric heterogeneous chemistry more than a year after the 2019/2020 wildfire event.

## 1 Introduction

During the austral summer of 2019–2020, wildfires across southeast Australia burned an approximate 250 thousand square kilometers. The most intense fires occurred in late December 2019 and early January 2020, known as the Australian New Year wildfires (ANY wildfires), with multiple pyrocumulonimbus (pyroCB) events injecting approximately 1 Tg of smoke into the stratosphere, with the majority raising to altitudes of ~22 km (Boone et al., 2020; Peterson et al., 2021), and additional wildfire-induced anticyclonic smoke vortices lofting as high as 35 km (Kablick et al., 2020; Khaykin et al., 2020; Lestrelin et al., 2021; Sellitto et al., 2022). The unprecedented amount of wildfire smoke increased lower stratospheric temperatures (Rieger et al., 2021; Yu et al., 2021), and affected stratospheric circulation, which played a role in dynamically controlling the extent of the Southern Hemisphere polar vortex (Damany-Pearce et al., 2022; Senf et al., 2023). The wildfire smoke also resulted in remarkable Southern Hemisphere midlatitude and polar chlorine processing. Significant HCl depletion was observed, maximizing in the winter around June/July of 2020, with most of the chlorine being partitioned into $ClONO_2$, and smaller amounts into active forms of chlorine like ClO (Bernath et al., 2022; Santee et al., 2022; Strahan et al., 2022). The HCl and $ClONO_2$ observed in the following winter of 2021 were less unusual but nonetheless outside the range of observed data for previous years since 2005. Solomon et al. (2023) showed that the likely cause of the unusual 2020 chemistry was high solubility of HCl in organics, which drove enhanced heterogeneous chemistry at warmer temperatures where typical background sulfate aerosols are too acidic for HCl to readily dissolve (Hanson et

al., 1994; Hanson & Ravishankara, 1994). These highly unusual composition changes have been replicated in model simulations covering the first 10 months after initial injection of the smoke. Here, we aim to investigate whether we can model smaller effects also seen in 2021 while also taking into consideration potential differences between wildfire and background organic aerosols.

Following the pyroCBs, significant enhancements of water and biomass burning products were detected in the Southern Hemisphere stratosphere (Schwartz et al., 2020). After injection, black carbon present within the smoke acted to "self-loft" the plume, with modelling studies suggesting the extent of the lofting was consistent with the smoke plume containing 2.5% black carbon (Yu et al., 2021). The aerosol extinction in 2021 remained elevated (Ohneiser et al., 2022; Santee et al., 2022). Further, Wang et al. (2023) showed through observations that abnormal chlorine chemical processing also occurred in 2021, suggesting that wildfire organics played a multi-year role in Southern Hemisphere lower stratospheric chemistry.

Nonvolcanic stratospheric sulfate aerosol mixing ratio peaks between approximately 20–25 km (Kremser et al., 2016), however, in the very lowermost stratosphere, background aerosols can be largely comprised of sulfate-organic mixtures (Murphy et al., 2014; Murphy et al., 2021), and thus organic material can contribute up to 40% of the total stratospheric aerosol optical depth (Yu et al., 2016). These "background organic" aerosols have noticeable differences from pyroCB sourced organic aerosols. For example, pyroCB aerosols have a larger mode size (Katich et al., 2023; Murphy et al., 2021).

Enhanced midlatitude heterogeneous chemistry after a volcanic eruption is dominated by $N_2O_5$ hydrolysis, the rate of which gets saturated as aerosol surface area density increases because $N_2O_5$ begins to get destroyed as fast as it can be created (Prather, 1992). Solomon et al. (2022) showed that $N_2O_5$ hydrolysis likely also occurred on the 2020 wildfire aerosols, causing characteristic $NO_x$ loss through conversion to $HNO_3$. Decker et al. (2024) showed that over the first few weeks after fires in North America, $N_2O_5$ reactive uptake values were smaller than typical in the upper troposphere and lower stratosphere. However, whether this remains the case on wildfire smoke as they age over longer time periods in the stratosphere is unknown. After the ANY wildfires, all heterogeneous chemistry reactions that are dependent on HCl solubility were likely also enhanced.

Measurements of how the presence of organics in lower stratospheric mixed aerosols (containing sulfate, organics, dust, etc.) affects the heterogeneous kinetics have not currently been reported in the literature. During 2020, the very large amount of organics allowed them to be considered independently when estimating the faster kinetics in organics due to greater HCl solubility (Solomon et al., 2023). However, for smaller concentrations of wildfire organics, such as in 2021, considerations need to be made on the morphology of the aerosols, for example, whether they may be homogeneously mixed with sulfate or exhibit liquid-liquid phase separation (LLPS) characteristics. Understanding of the processes that govern LLPS in mixed aerosols in the atmosphere has increased significantly in recent years. There is evidence that mixed aerosols containing organic and inorganic species display some form of LLPS, either with an organic core-shell, or in a partially engulfed morphology (Freedman et al., 2024 and references therein). Some examples of drivers that act to control the phase separation are: the acidity of the organic-inorganic mixture (Tong et al., 2022), the size of the aerosol, with smaller sizes being more likely to limit LLPS (Veghte et al., 2013), the temperature and viscosity (You and Bertram, 2015), and the oxidation state, (expressed as O:C ratio) of the organics which changes their polarity and hydrophilicity (Gorkowski et al., 2020). A less oxidative state is more likely to form a partially engulfed or core-shell LLPS, and a more oxidative state tends to form a more homogeneous mixture (Gorkowski et al., 2020). These drivers are important for consideration of the form of the aerosols as they are emitted and age in the atmosphere (Marcolli and Krieger, 2020). Considering the complexity of the parameters that appear to influence LLPS, background organic aerosols could have different morphologies

and different mixed states compared to pyroCB sourced organics, especially given the observed differences in diameter between background and wildfire organic aerosols (Katich et al., 2023; Murphy et al., 2021). There is also evidence that organic aerosols can become glassy (e.g. Virtanen et al., 2010), however, similar to previous work by Solomon et al. (2023), here we assume the particles are liquid. Other considerations, such as differences that eucalyptus sourced organics from the ANY wildfires could have compared to Northern Hemisphere wildfires are not investigated here but could contribute to the large chemical perturbations seen in 2020.

In this paper, we investigate and model the multi-year role of 2020 wildfire organics on stratospheric chemistry in Southern Hemisphere midlatitudes and polar regions. We discuss how this chemistry will evolve assuming both wildfire and background organic sources act similarly (for example, if the organics are homogeneously mixed with sulfate or form LLPS aerosols) or if they are different. The seasonality of the chemistry perturbations is also discussed.

## 2 Data and Methods

### 2.1 Observations

We use the MLS Level 2, version 5 PressureZM (zonal mean values on pressure levels) measurement product for ozone, ClO, and HCl data to analyze midlatitude (40–55°S) and polar (70–80°S) time series. Note, it is recommended to use the MLS ClO day products minus night products in the midlatitudes to avoid biases where feasible (Livesey et al., 2022). However, since we are primarily analyzing anomalies, and to better compare with our modeled daily averaged ClO values, the MLS ClO daily average values are used here. We also use the Atmosphere Chemistry Experiment-Fourier Transform Spectrometer (ACE-FTS) version 5.2 data for $ClONO_2$ and HCl over the same pressure levels and midlatitude and polar regions as for MLS (Bernath et al., 2005; Boone et al., 2023). The use of ACE-FTS HCl data here gives a set of observations independent of MLS to test against our model results. ACE-FTS has sporadic spatial coverage for specific latitude ranges. Therefore, monthly averages of the available daily data for each month are constructed. ACE-FTS quality control is performed by removing data points that lie outside three standard deviations of the mean values, as suggested for this data product. The Ozone Mapping and Profiler Suite Limb Profiler (OMPS-LP) measurements of aerosol extinction at 675 nm using the retrieval method of the NASA Goddard space flight center are also used to track the elevated smoke through 2021 and compare with model simulated aerosol extinction at the same wavelength (Taha et al., 2021).

### 2.2 CARMA-CESM1

Similarly to Solomon et al. (2023), this study uses the Community Earth System Model (CESM1) coupled with the sectional aerosol model CARMA (Bardeen et al., 2008; Toon et al., 1988; Yu et al., 2015, 2016). The model is spun-up in whole atmosphere specified dynamics mode from midsummer 2019 until 29 December 2019 using MERRA2 reanalysis of winds and temperatures (Gelaro et al., 2017). The run is then started as a free-running simulation on 29 December 2019, when the first pyroCB event occurred. Emissions of ozone depleting substances are from CMIP5 (Meinshausen et al., 2011). In the southern hemisphere midlatitudes, the model total column inorganic chlorine ($Cl_y$) is between $4 \times 10^{15}$ and $5 \times 10^{15}$ molecules/cm$^2$, and inorganic bromine ($Br_y$) is between 19-21 ppt in good agreement with the observed and inferred values (WMO, 2022). We inject a total of 0.9 Tg of smoke over southeastern Australia (39° S, 150° E) at ~12 km on days that experienced large pyroCB events (29-31 December 2019 and 4 January 2020). As we begin the simulation in free running mode, the smoke can self-loft due to the inclusion of 2.5% black

carbon which was shown by to compare well with the observed amount of self-lofting. However, the model does not simulate the anticyclonic vortices that put some aerosol into the middle stratosphere. On March 1, the model is switched to specified dynamics mode to appropriately capture the future transport of the smoke in the stratosphere using MERRA2 reanalysis. The model has a horizontal resolution of 1.9° latitude × 2.5° longitude and 56 vertical layers with a model top at 1.8 hPa.

The model calculates primary organic aerosols (such as organics emitted directly into the atmosphere through the wildfires) in both mixed aerosols and primary organic sections. The primary organic section only contains primary organics, while the mixed section contains a mixture of sulfate, organics, black carbon, sea salt, and dust. Both sets of particles containing primary organics are computed in 20 size bins whose radii range from 0.05 to 8.7 μm. 80% of the organic portion of the smoke is injected into the pure organic bins and the remainder is injected into the mixed aerosol bins with black carbon. The organics then age by a prescribed effective reaction probability of $10^{-6}$ that encompasses oxidation by ozone or OH and photolysis for the organic portion (Yu et al., 2019). Condensation of sulfuric acid from the gas phase and coagulation of pure sulfate onto the increased surface area of mixed aerosols also can occur in the model. Secondary Organic Aerosols (SOA) using the volatile organic compound precursors: isoprene, monoterprene, benzene, xylene, and toluene are also included but instead are treated as bulk aerosol with the assumption that they are contained in mixed aerosols for calculation of certain aerosol properties, such as size distribution (Yu et al., 2015). Pure sulfate bins are also included.

Heterogeneous chemistry reactions in stratospheric aerosols that are important for this study are

$$ClONO_2 + H_2O \rightarrow HNO_3 + HOCl \qquad \text{(R1)}$$
$$ClONO_2 + HCl \rightarrow Cl_2 + HNO_3 \qquad \text{(R2)}$$
$$HOCl + HCl \rightarrow H_2O + Cl_2 \qquad \text{(R3)}$$
$$HOBr + HCl \rightarrow BrCl + H_2O \qquad \text{(R4)}$$

Reactions R1-R3 follow Shi et al. (2001), and reaction R4 follows Hanson (2003) and Waschewsky & Abbatt (1999). Hanson, (2003) and Waschewsky and Abbatt (1999) reported different measurements of the solubility of HOBr in sulfuric acid solutions, resulting in different values for the second order rate constant for reaction R4; this model uses Hanson (2003) solubility values and Waschewsky and Abbatt (1999) second order rates that have been adjusted to agree with the Hanson data, as such it should be noted that the reaction R4 rates are likely an upper limit (e.g. Zhang et al., 2024). However, reaction R4 plays a secondary role to reaction R2.

**2.3 Considerations on HCl solubility**

HCl solubility is a key parameter for stratospheric heterogeneous kinetics on liquid aerosols. All heterogeneous reactions that involve HCl are dependent on the molarity of HCl in the particle. Typical sulfate aerosols are too acidic for significant HCl solubility at midlatitude temperatures, limiting the probabilities of these reactions. Solomon et al. (2023) used laboratory measurements that obtained large HCl solubility in oxidized organic species to represent HCl solubility in stratospheric organic aerosols from the ANY wildfires. This approach resulted in enhanced heterogeneous chemistry in good agreement with the 2020 observations. That study employed enhanced solubility anywhere that a wildfire organic to sulfate mass ratio exceeded 1. This is appropriate for very large wildfire organic concentrations, as seen in 2020, but is not suitable for investigating the smaller

concentrations in 2021 that have likely combined with sulfate and formed homogeneous or LLPS mixtures, and for investigating whether background organics also act similarly to the wildfire aerosols.

While the exact morphology of the aerosols is unknown without further observations, here we present three cases to parse the efficacy of ANY wildfire organics and background organics in dissolving HCl in 2021 with the following assumptions and outcomes described below.

1) Background and ANY wildfire organics dissolve HCl similarly and are homogenously mixed with sulfate in the model mixed bins, while the model pure sulfate bins remain separate. The amount of background organic aerosol in the lower stratosphere is considerable, ~40% of total stratospheric aerosol optical depth (Yu et al., 2016). Our version of CESM1-CARMA has background organics being virtually all SOA (see Figure S2). Here we assume wildfire and background organics are homogeneously mixed with sulfate such that the solubility of HCl is affected by the sulfate acidity within the mixture,

$$H^* = H(1+Ka/aH) \tag{1}$$

Where $H^*$ is the effective Henry's law constant, Ka is the dissociation constant of HCl, as used in Solomon et al. (2023), and aH is the acidity calculated from an adjusted $H_2SO_4$ weight percent that accounts for the addition of organic mass (Shi et al., 2001). The mixed aerosol organic mass is taken as the combined mass of SOA plus the mass of all ANY wildfire organics (taken as a sum of mass from all 20 size bins). Note that organic molecular weight is not needed to adjust the $H_2SO_4$ weight percent. This allows the investigation of the ANY wildfires effect on chemistry assuming that background and wildfire organics act identically within a homogeneous mixture with sulfate. The solubility in mixed aerosols is then linearized by the fraction of organics and sulfate, where the solubility of HCl in the organic portion proceeds by equation 1 after accounting for the mole fraction within the mixed aerosol, and the HCl solubility in the sulfate fraction proceeds at the standard rate as described in Shi et al. (2001). The solubility in the pure sulfate aerosols also proceeds at the standard rate. Note, this method does not consider any differing size distributions of wildfire and background aerosols, any differences in morphology or phase, or any differences in SOA and wildfire organics in their ability to dissolve HCl. The goal of this approach is to determine if background organics act the same as wildfire organics in a homogeneous mixture, and if this consideration would still be in good agreement with observations over time as the wildfire organics are slowly removed.

Figure S1e and S1f show the results of case 1 (homogenized wildfire and homogenized control) compared to MLS at 68 hPa as daily anomalies from a control run for the model (where only SOA is considered when assuming homogenized mixed aerosols for HCl solubility) and from the observed daily mean climatology (2004–2019) for MLS. We obtain excellent agreement in 2020, however in 2021, when stratospheric wildfire organics loadings are significantly lower, we do not see good agreement with observations. In fact, we see that the control run causes greater heterogeneous loss compared to the wildfire run (see Figure S1f for anomalies from respective control). This is occurring because of coagulation of pure sulfate aerosols and condensation of gas phase $H_2SO_4$ which increases both the total sulfate and mixed sulfate in the wildfire run compared to the control run (Figures S1a-d). For 2020 this did not have much effect because wildfire organic concentrations were so high. However, in 2021, as wildfire organic concentrations continue to diminish, the extra mixed sulfate causes the organic to sulfate mass ratio to be lower in the wildfire run compared to the control run. This results in a more acidic mixed particle in the wildfire case compared to the control

case and therefore lower HCl solubility. This is an indication that wildfire and background SOA are not homogeneously mixed in such a way that acidity is important for controlling HCl solubility over time.


2) Background organics act similarly to wildfire organics but are not homogeneously mixed; rather, organics and sulfate display LLPS. This case is an LLPS control run that does not include the ANY wildfire. We parameterize LLPS through the assumed effects it will have on HCl solubility in background SOA (see Figure S2). As the organics are no longer assumed to be homogeneously mixed, they will not be affected by the acidity of sulfate. Therefore, in this case, HCl solubility is linearized by

the mass fraction of background organics to sulfate in mixed aerosols, while solubility in pure sulfate aerosols remains separate. It is noted that when assuming LLPS, a more realistic approach would be linearizing by the surface area of sulfate and organics in the mixed aerosols rather than the mass. This would also depend upon whether the LLPS is a core-shell or partially engulfed morphology. However, given that the exact morphology of these particles is unknown, the simplest approach is to linearize by mass fraction, as is done here, with the acknowledged caveat that this may be an underestimation if the characteristics of LLPS are

indeed an organic core-shell morphology. Additionally, If we compare Figure S2 to what is known about background aerosols, the model overestimates the amount of background SOA, especially in the Southern Hemisphere, by approximately 4 times as shown in Table 2 in Murphy et al. (2021), where a volume fraction of organic-sulfate mixtures in total aerosol volume ranging from approximately 0.1–0.4 was observed in the lower stratosphere depending on the Atmospheric TOMography mission. For mixed aerosols, the solubility of HCl in the organics fraction follows Solomon et al. (2023), which used laboratory measurements in

hexanoic acid as a proxy for organic species. Solubility in the sulfate fraction proceeds at the regular rate (Shi et al., 2001).

Figures S1e and f show the results of this simulation compared to MLS HCl (LLPS background SOA). As can be seen, if background organics are assumed to be LLPS and act similarly to wildfire organics, they deplete HCl in austral winter in the Southern Hemisphere midlatitudes in a manner that is inconsistent with MLS, i.e. HCl has too low a minimum in austral spring.

However, as indicated previously, the model overestimates background SOA in the Southern Hemisphere midlatitudes. Dividing the background SOA organic fraction by 4 brings the results closer to observations, but with similar inconsistencies (Figure S1e and f, LLPS background SOA/4). Another important thing to note that could introduce uncertainties in the model representation of HCl and therefore the background SOA chemistry is an uncertainty in very short lived substances (VSLS) trends (Chipperfield et al., 2020; Hossaini et al., 2019), although transport effects such as uncertainties in the mean circulation in MERRA2 also could

contribute. Therefore, we can't rule out that background SOA is also enhancing heterogeneous chemistry in the Southern Hemisphere, although it seems likely to be occurring at a slower rate compared to the ANY wildfire aerosols based on the present simulations.

3) We investigate whether only the ANY wildfire organics are consistent with a LLPS assumption, and dissolve HCl at a higher

rate than sulfate aerosols. This approach uses the same methodology as case 2 for HCl solubility. Background SOA is assumed to have a similar efficacy to sulfate in dissolving HCl and is therefore considered in the background aerosol mass when constructing the ANY organic mass fractions. Through this approach it is assumed that HCl solubility will remain elevated as long as there are excess wildfire organic aerosols present. Therefore, changes in enhanced heterogeneous rates over time will scale with the transport of the smoke, surface area density, aerosol radius, wildfire organics oxidation, and sedimentation of Australian wildfire organic

aerosols. Note that this method is most similar to what was presented in Solomon et al. (2023), but differs through the linearization of organics to background aerosols, as outlined above. The results of this method are presented in Figures 1-4.

## 3 Results

### 3.1 Stratospheric aerosol extinction anomalies

Figure 1 shows 675 nm aerosol extinction daily mean anomalies (difference of each day from daily mean climatologies) for OMPS-LP and CESM1-CARMA for both Southern Hemisphere midlatitude (55–40°S) and polar regions (80–70°S). The CESM1-CARMA anomaly is the difference from the control run. The OMPS-LP anomaly is the difference from years that were volcanically clean (2012, 2013, 2014, 2017). Volcanic eruptions that occurred in 2019, 2020, and 2021 that also increased the extinction in the observations likely include Ulawun in June and August 2019, Taal in January 2020, and La Soufrière in April 2021 (Asher et al., 2024; Yook et al., 2022). Compared to OMPS-LP, CESM1-CARMA has slightly lower extinction anomalies during the end of 2020 and into 2021 for the midlatitudes at 18.5 km which could be due to enhancement in aerosol extinction from volcanic sources which is removed in the model anomalies as both the control and wildfire runs include volcanoes. For example, the influence of Ulawun can be seen in late 2019 in Figure 1 where OMPS-LP extinction is elevated before the ANY wildfires. The influence of La Soufrière can be seen in July 2021 in Figure 1a, and 1b where OMPS-LP extinction levels start increasing again in contrast to the model. Even though the model shows lower extinction anomalies, they do remain elevated into 2021 at both 18.5 km and 16.5 km. In the polar regions the model does not capture the immediate increase in extinction that is seen in OMPS-LP. However, it should be noted that January-February in CESM1-CARMA uses free-running dynamics to ensure that the smoke plume is able to self-loft. Therefore, the initial transport to the polar regions will be subject to free running variability. After February CESM1-CARMA does have elevated aerosols, so polar effects can be simulated. Throughout the late winter and summer, the aerosols decrease significantly as they sediment out of the stratosphere. Once the polar vortex weakens, elevated aerosols from lower latitudes are transported into the polar region during January-March of 2021. This is seen in both OMPS-LP and CESM1-CARMA, and is very clearly associated with the transition of 10 hPa zonal winds from westerlies to easterlies (shown as the vertical brown dotted line from MERRA2 reanalysis). OMPS and CESM1-CARMA show excellent agreement in this regard for both 16.5 and 18.5 km, highlighting that the polar regions contained significantly elevated aerosols that contained wildfire smoke in early 2021. A complementary figure showing the ratio of 2020–2021 extinction to background values is shown in Figure S3, with both midlatitude and polar 2021 values ranging between ~1.5 times background values at 18.5 km during January-April and approaching 2 times background values at 16.5 km.

**675 nm aerosol extinction anomaly**

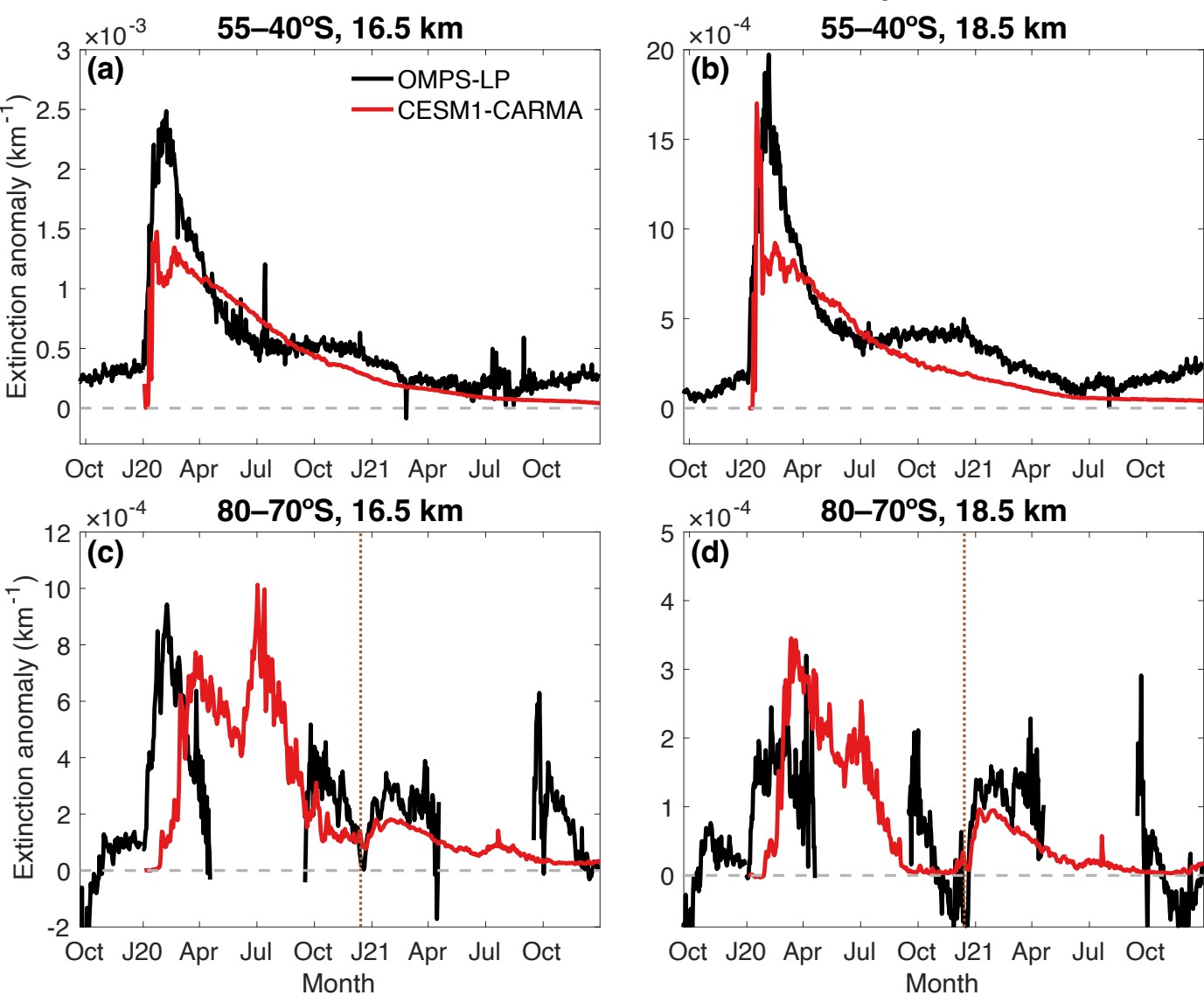

Figure 1. Comparison of OMPS-LP and CESM1-CARMA 675 nm aerosol extinction coefficient anomalies for southern midlatitudes (a, and b) and polar regions (c, and d). The OMPS-LP anomaly is the difference from volcanically clean years. i.e. 2012, 2013, 2014, and 2017. The CESM1-CARMA anomaly is the difference from the control. Vertical dotted line in c, and d represents the date that 10 hPa zonal wind transition from westerly winds to easterly winds in MERRA2.

**3.2 Midlatitude chlorine partitioning**

Figure 2 shows Southern Hemisphere midlatitude HCl, ClONO$_2$, ClO, and ozone daily mean anomalies at 68 hPa and 100 hPa over 2020 and 2021 for CARMA-CESM1, MLS, and monthly mean anomalies for ACE-FTS. The anomalies for CESM1-CARMA are the difference from the control while the anomalies for the observations are the difference from MLS or ACE-FTS respective climatologies over 2004–2019. See figure S4 for absolute values. At 68 hPa we obtain excellent agreement between the model and observations for all species. The majority of Cl from HCl loss is partitioned into ClONO$_2$, with a smaller amount going into ClO (Figure 2e) and other species, such as HOCl (Bernath et al., 2022; Solomon et al., 2023). The record amount of active chlorine in midlatitudes results in around 0.2 ppm loss of ozone (~10-15%) in 2020 in the lower stratosphere. The HCl solubility linearization

method used here allows for appropriate simulation and comparison of the wildfire organics into 2021. Using this method, we find that HCl recovers close to climatology levels by December 2020 in agreement with MLS and ACE-FTS, before decreasing again to levels that are outside the MLS 2004–2019 climatology by July 2021, and subsequently recovers again by the end of 2021. The HCl loss in 2021 is again mirrored by $ClONO_2$ increases. The Cl partitioning in 2021 only produces small ClO anomalies in the model, while indiscernible changes in the amounts of ClO are obtained in the observations. However, ozone doesn't fully recover to control run levels from the loss in 2020 until the end of 2021, which is in reasonable agreement with observations.

Model results at 100 hPa also show similar chlorine partitioning compared to 68 hPa; however, the model produces too much HCl loss compared to observations. The cause of this is unknown. One option could be due to the organics becoming more viscous or glassy at colder temperatures near the tropopause, which would limit the uptake of HCl. Model uncertainties due to VSLS could also play a role in comparing with observations at lower stratospheric altitudes (Hossaini et al., 2019). However, even with these considerations and the differences compared to the model, the MLS HCl at 100 hPa is still outside the 2004–2019 climatology in July 2021 and shows good agreement with ACE-FTS, and the timing of the loss in the model suggests that abnormal chemistry is also occurring in 2021 at 100 hPa with similar seasonality to that of 68 hPa.

The good agreement with observations in 2021, especially at 68 hPa suggests that the effectiveness of ANY wildfire organic aerosols to dissolve HCl does not change significantly over 2020–2021. The smaller effect in 2021 compared to 2020 is therefore due to the decrease in the amount of aerosol surface area and the amount of wildfire organics remaining in 2021, (expressed here as a lower organic to sulfate mass fraction).

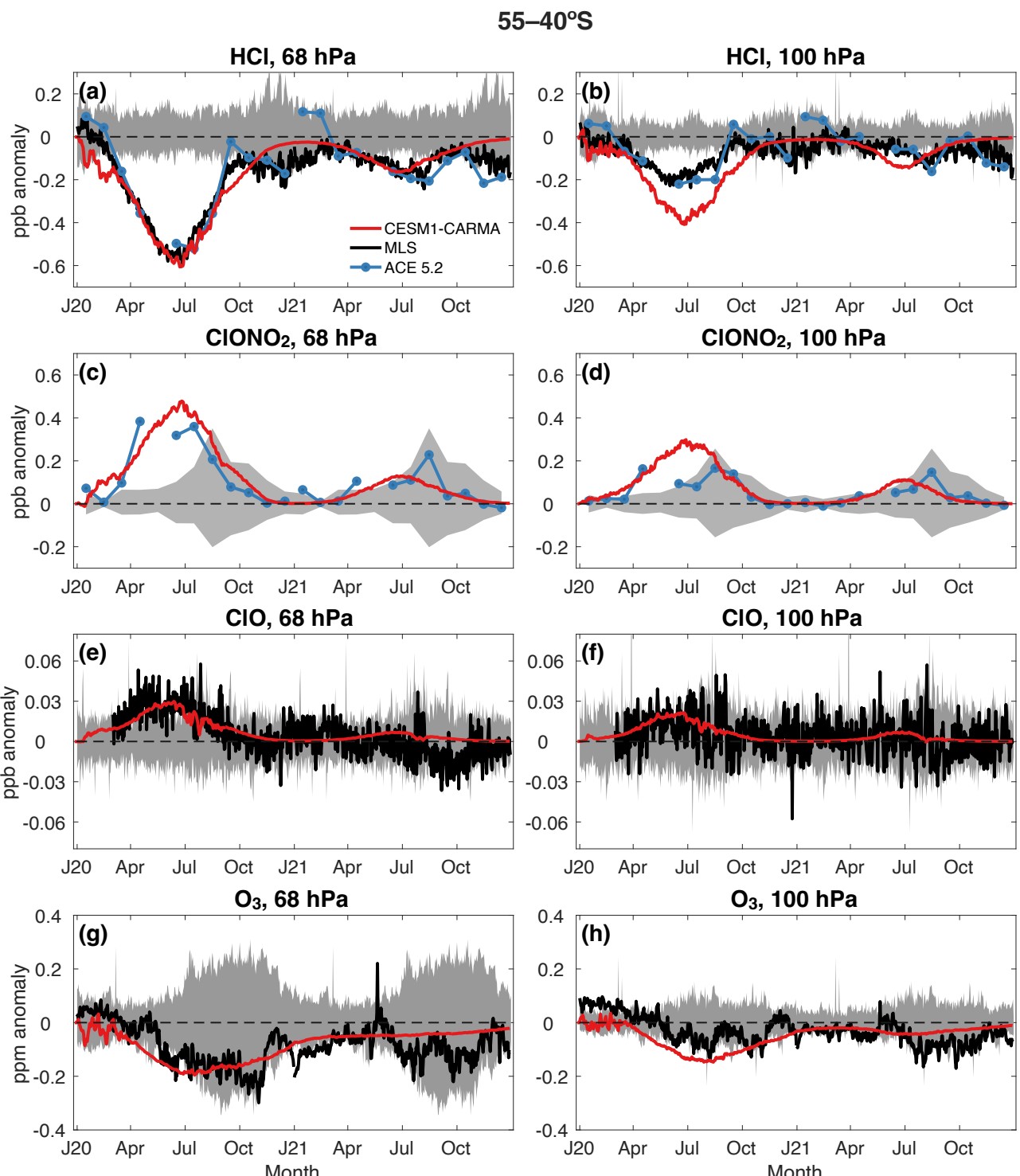

Figure 2. Comparison of midlatitude (55–40°S) HCl, ClONO2, and O3 anomalies between MLS and ACE-FTS observations and CESM1-CARMA model results. Anomalies are daily mean anomalies for CESM1-CARMA and MLS and monthly mean anomalies for ACE-FTS. MLS and ACE-FTS anomalies are differences from their respective climatologies (2004–2019). CESM1-CARMA anomalies are the differences from the control. The grey shading shows the MLS or ACE-FTS variability.

### 3.3 Seasonality and vertical structure

Figure 2 shows a clear seasonality in the Cl partitioning due to the organic enhanced heterogeneous chemistry. In both 2020 and 2021, especially at 68 hPa, modeled HCl/ClONO2 loss/production peaks in austral winter before recovering close to the control

run values. This is also observed in MLS and ACE-FTS measurements. This seasonality and its vertical structure become clearer in Figure 3b, where the difference between the wildfire run and control run removes the role of dynamical variability due to the specified dynamics model setup.


This seasonality appears to occur independently of wildfire organic aerosol concentration, as there are more organic aerosols (as seen by the higher midlatitude extinction anomaly in Figure 1) in the summer of 2020/2021 compared to the winter of 2021. It is also independent of temperature, as our increased heterogeneous reactions on organic aerosols are relatively insensitive to temperature. Nor is it a dynamical phenomenon. (All three of these considerations were confirmed in a photochemical box model

with no transport, where organic mass concentration, temperature, surface area density, aerosol radius, $H_2O$, and $CH_4$ where kept constant). Therefore, this seasonality is a photochemical phenomenon. We find that it is dependent on both the seasonality of photolysis rates and daylength, and their resulting control on short lived species that are important for $ClO_x$ partitioning, such as NO and $NO_2$.

The heterogeneous reactions on organic aerosols are somewhat temperature independent, and therefore don't vary greatly over the course of the year. This produces a consistent source of extra activated chlorine, primarily in the form of $Cl_2$ through reaction R2. $Cl_2$ from reaction R2 will rapidly photolyze, producing 2Cl, which will then rapidly react with ozone to form ClO. How fast ClO is converted back to Cl is dependent on reaction R5, a primary coupling reaction that links the ClOx and NOx cycles. As this coupling reaction slows down in winter due to the seasonality of NO, the Cl/ClO ratio decreases (as shown in figure S5b).

Additionally, the daytime amount of $NO_2$, which is important for reaction R6, has a smaller seasonal amplitude than NO, resulting in reaction R6 partitioning more of the extra ClO into $ClONO_2$ in winter compared to summer (see Figure S5c, d, and e) so that more relative HCl is created though reaction R7. This in combination with a daylength controlled $ClONO_2$ photolysis rate, produces the observed and modeled seasonality.

$ClO + NO \rightarrow NO_2 + Cl$ (R5)

$ClO + NO_2 + M \rightarrow ClONO_2$ (R6)

$Cl + CH_4 \rightarrow HCl + CH_3$ (R7)

Figure 3 also highlights the vertical structure of the anomalies in the Southern Hemisphere midlatitudes for HCl and ozone. Both

the model and observations show a peak HCl loss occurring at approximately 68 hPa in both 2020 and 2021 in the austral winter (Figure 3a, and 3b). However, the model HCl anomalies are peaking at slightly higher pressure compared to the observations. In 2021 the model anomalies extend up to approximately 30 hPa, while the only MLS anomalies that are distinguishable from MLS 2004-2019 variability are occurring between 70 and 100 hPa in late July and August. Figures 3c, and 3d show ozone anomalies for MLS and the model respectively. Similarly to HCl, the model ozone peak anomaly is occurring at slightly higher pressure

compared to MLS. However, the modeled timing and temporal extent of negative ozone anomalies in the lower stratosphere agree very well with MLS. Peak chemical ozone loss in 2020 is occurring between July and October (but only times in late May and June show anomalies outside the MLS climatology). The model shows a second ozone loss peak in 2021 at higher pressure between July and October, in agreement with the timing of 2021 negative anomalies seen in MLS in the lower stratosphere. Therefore, the comparison between the model and the data also suggests that the ANY wildfires contributed to the record low ozone seen in MLS

in 2021 at altitudes below 100 hPa.

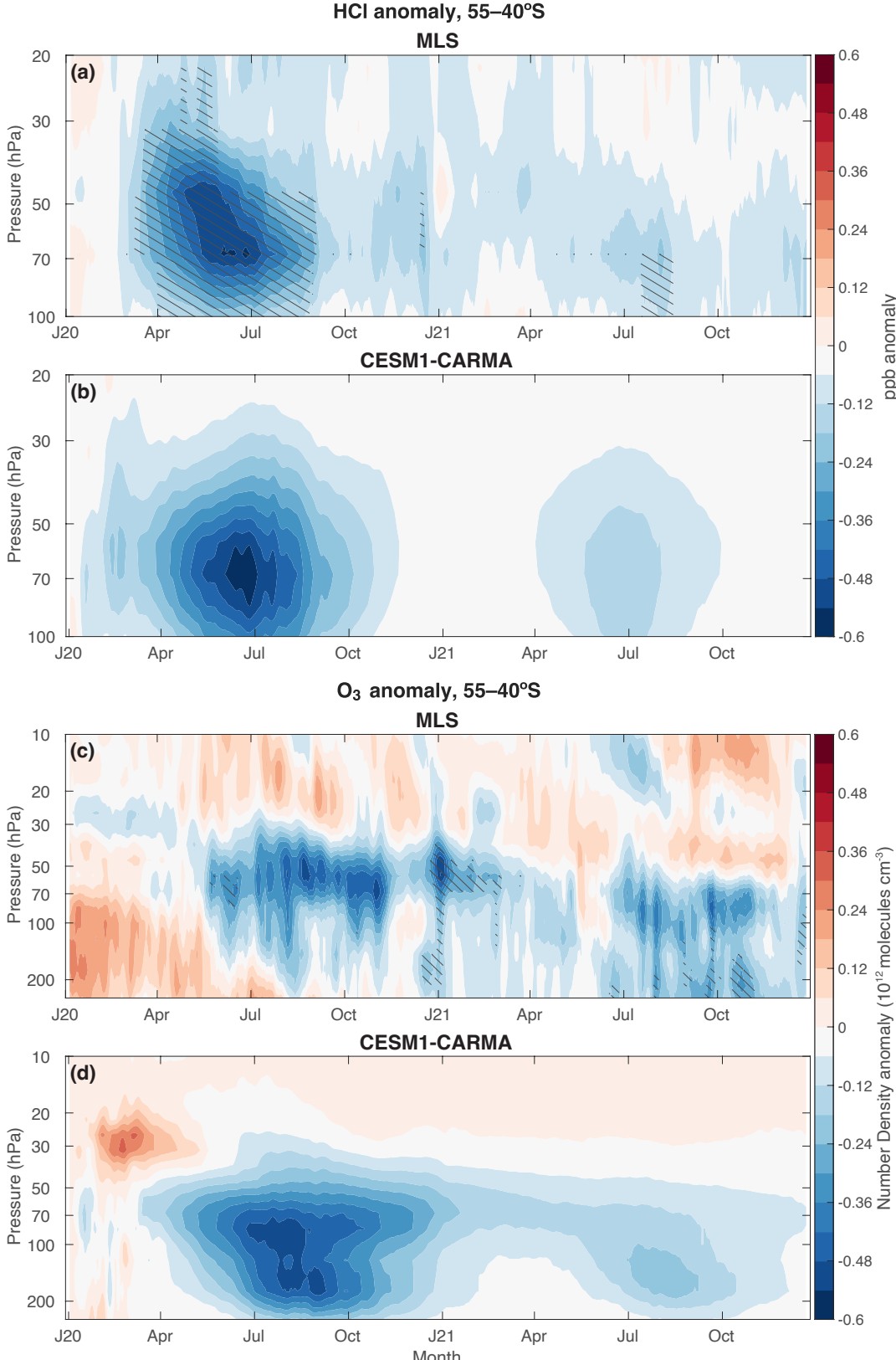

Figure 3. a) MLS 2020–2021 55–40°S HCl volume mixing ratio anomaly from the MLS 2004–2019 mean. b) CESM1-CARMA modeled HCl anomaly from control. c) MLS 2020–2021 ozone number density anomaly MLS 2004–2019 mean. d) CESM1-CARMA modeled ozone anomaly from control. All data has been smoothed by a 7-day running average. Hatched regions show values that lie outside the MLS variability over 2004–2019.

**3.4 Polar chlorine partitioning**

In the Southern Hemisphere stratospheric polar winter, the isolation of the vortex and absence of sunlight allows for almost complete conversion of the chlorine reservoir species HCl and ClONO$_2$ to Cl$_2$ through reaction R2 on polar stratospheric clouds. The Cl$_2$ subsequently gets converted to atomic Cl during austral spring when the sun rises, catalytically destroying ozone. The observed winter conversion of the reservoir species is shown in Figure 4a, and 4b, where we see near complete loss of HCl by the beginning of July in MLS. Typically, chemistry-climate models can capture the timing of the loss of ClONO$_2$, but complete HCl loss does not occur until late August in the deep vortex in models, highlighted in the control run in Figure 4a and 4b. The mechanism for this model anomaly is currently unknown. One potential reason is a discrepancy in the availability of ClONO$_2$ in the deep vortex to continue to drive reaction R2 (Grooß et al., 2018). In contrast to Figure 2, we show absolute values in figure 4 to be able to properly compare the timing of HCl loss and to compare the size of the ozone holes between the model and the observations. Showing absolute values indicates there is a negative bias in modelled HCl and a positive bias in modelled ClONO$_2$. This results in modelled HCl not recovering to similar levels in October 2020 and 2021 as seen in observations. However, these biases are consistent with and without the inclusion of organics.

During 2020 and 2021, while wildfire organic aerosols were elevated, HCl was completely removed from the deep vortex up to a month earlier than is typically observed at 68 hPa and up to 2 months earlier at 100 hPa as shown in Figure 4a, and 4b (See Figure S6 for anomalies), and this is also captured by the model, due to excess ClONO$_2$ that was created earlier in the year through enhanced heterogeneous chemistry on wildfire organic aerosols when sunlight was still available to produce NO$_2$, and before the denitrification of the Antarctic stratosphere (Figures 4c, and d). The early enhanced heterogeneous chemistry is accompanied by a modeled elevation of ClO anomalies at 68 hPa during the first 5 months of 2020 that is in good agreement with the observations (see Figure S6). There are small decreases in ozone in the model in both 2020 and 2021 early in the year. For the austral spring, since the model uses specified dynamics for both the control and wildfire cases, only direct chemical effects are captured. Thus, analyzing differences in this way cannot distinguish any chemical-radiative feedback, but since there is virtually no difference in austral spring ozone in the control run compared to the wildfire run, it is clear that an abnormally stable and persistent vortex during both 2020 and 2021 contributed to the large and prolonged springtime ozone loss. This is seen in the MLS observations which show large differences from MLS climatology from October to December in 2020 and 2021 but good agreement with the model. Additionally, dynamical influences from elevated wildfire organics may play a role in stabilizing the vortex (Damany-Pearce et al., 2022). Any chemical-dynamical feedbacks have not been analyzed here but could also play a role.

Over Antarctica, the largest effects of the wildfires on chemistry in 2021 are clearly seen at 100 hPa, with observed HCl and ClONO$_2$ still considerably outside the range of values observed between 2004–2019. These differences are also seen in the model, but not quite to the same extent. Considering that the 2021 extinction level anomalies are around 4 times less compared to 2020 at 16.5 km (Figure 1c), this indicates that lower organic concentrations have the potential to substantially perturb the timing of conversion of chlorine reservoir species in the polar region.

## 80–70°S

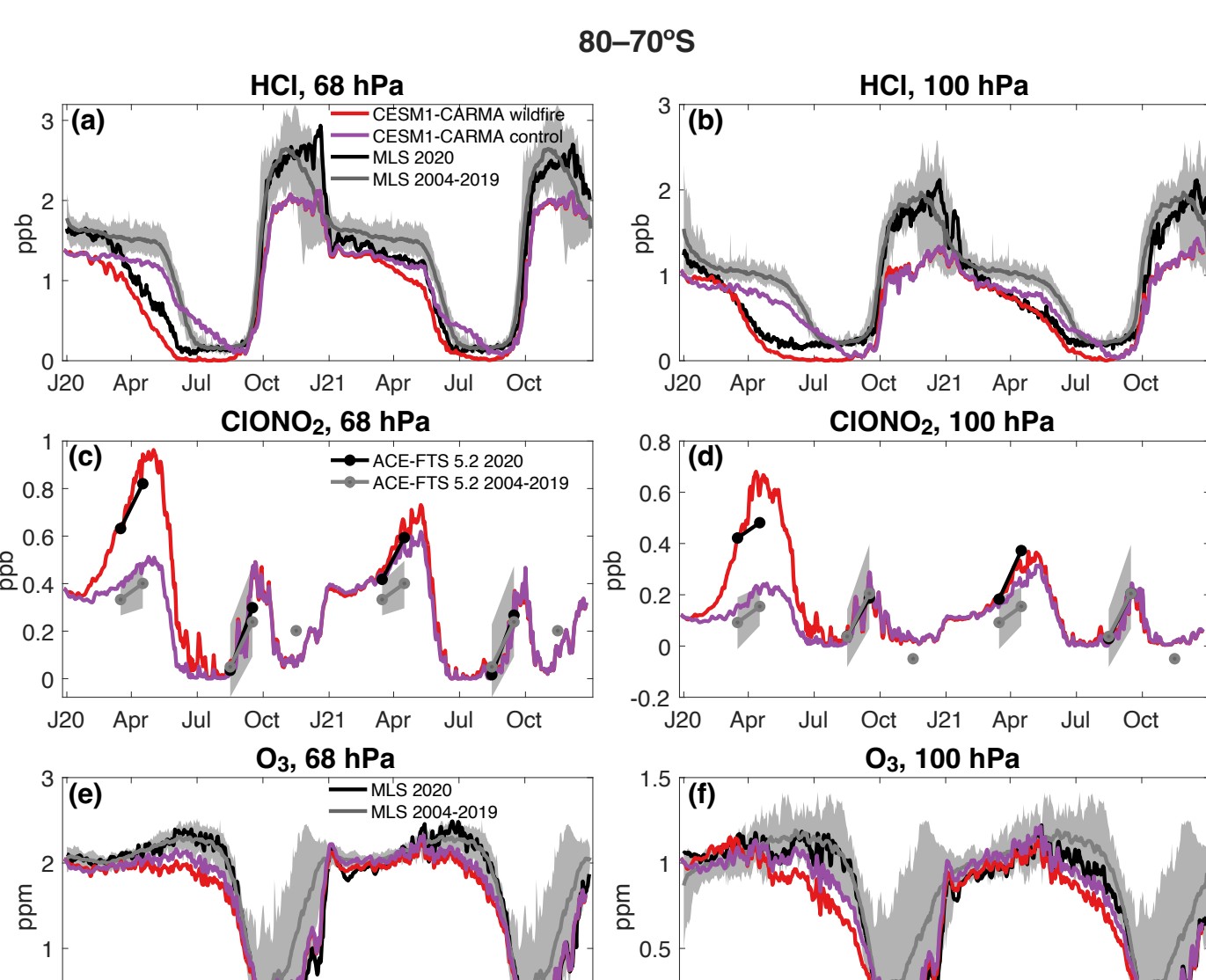

Figure 4. Comparison of the south polar region (80–70°S) HCl, ClONO₂, and O₃ absolute values between MLS and ACE-FTS observations and CESM1-CARMA model results for both 68 hPa (a, c, and d) and 100 hPa (b, d, and e). The grey shading shows the MLS or ACE-FTS variability.


## 4 Conclusions

In this work, the 2019–2020 Australian New Year wildfires effect on lower stratosphere chemistry has been shown to have a multiyear effect, highlighting that lower levels of wildfire organics can still substantially perturb chlorine partitioning in both the Southern Hemisphere midlatitudes and polar regions in 2021, up to 18 months after initial injection into the stratosphere through pyrocumulonimbus events, in good agreement with Wang et al. (2023). This is likely due to the aerosols maintaining their ability to dissolve HCl over time, and therefore the reaction rates scale with surface area density and organic concentration. This effect in 2021 is clearly seen in MLS and ACE-FTS observations, with decreases in HCl and increases in ClONO₂ and ClO occurring due to enhanced heterogeneous chemistry. The largest differences were observed at 68 hPa during July–August for Southern Hemisphere midlatitudes and April–May at 100 hPa in the polar region. Modeled midlatitude ozone depletion is in good agreement


with observed MLS ozone anomalies, with ozone depletion in 2021 peaking in the austral winter, indicating that the ANY wildfires likely contributed to the record low ozone values seen in the lower Southern Hemisphere midlatitude stratosphere in 2021.


As described in Solomon et al. (2023), the enhanced heterogeneous chemistry is likely occurring due to HCl being highly soluble in organic species. Here, we expand on this work by constructing a linearization of the solubility of HCl based on the wildfire organic to sulfate ratio in stratospheric aerosols. While the exact morphology of the aerosols is unknown, model tests suggest that this linearization applies to the aerosols as they underwent liquid-liquid phase separation, coagulating and combining with

background aerosols in the stratosphere. This linearization technique produces good results in comparison with observations for Southern Hemisphere midlatitude and polar regions over 2020–2021 in the lower stratosphere. Therefore, it is probable that the capacity of the ANY wildfire organic component to dissolve HCl does not largely change over a 2-year timescale, and thus the heterogeneous rates are primarily driven by surface area and organic concentrations.

The model overestimates Southern Hemisphere background SOA which results in too much HCl loss in a liquid-liquid phase separation control run without ANY wildfire smoke. Reducing the background SOA mass fraction in the linearization to more realistic levels for the Southern Hemisphere improves the comparison with observed background HCl at 68 hPa but it appears that the aerosols still drive too much heterogeneous HCl chemistry. However, due to uncertainties such as in the morphology of lower stratosphere background SOA (e.g., core shell vs partially engulfed), and the amount of VSLS chlorine, the extent that background

SOA acts similarly to ANY wildfire organics requires further investigation.

There is clear seasonality of the heterogeneous effects of wildfire organics on midlatitude stratospheric chemistry. A minimum in the HCl perturbation is occurring in winter. This in turn produces the most ozone loss in winter and spring in both 2020 and 2021 in the model, in good agreement with observations. This could have implications for enhanced ultra-violet exposure due to ozone

depletion if similar future events occur. The seasonality is driven by the photochemically controlled partitioning of ClO and Cl through important Cl coupling and reservoir reactions.

In the south polar region, increased $ClONO_2$ levels observed in the fall months before typical $ClONO_2$ loss allows for notably earlier complete loss of HCl with wildfire smoke than what is typically seen. This occurs in both 2020 and 2021, emphasizing that

lower wildfire organic concentrations can also cause large perturbations from the mean state in the polar region, even larger than that of midlatitudes. The excess $ClONO_2$ not only also enables the model to accurately replicate the complete HCl loss that is observed in 2020 and 2021, but also prompts this to occur up to 3 months earlier than is typically seen in chemistry-climate models due to the known discrepancy in the calculated polar HCl loss in standard models.

The agreement of the model with observations presented here is consistent with our assumption that the Australian wildfire organics are in a LLPS state. However, their exact morphology is not known, and more accurate heterogeneous kinetics of both wildfire and background organics are required to improve wildfire simulations. Additionally, the potential uniqueness of stratospheric organic aerosols from eucalyptus sourced pyroCB events that are typical of Australia, to that of pyroCB events from other regions of the world is unknown and requires further investigation. The results presented here are important for considering the effect of future

wildfire smoke injections through pyroCB events.

**Code Availability**

The model used in this study can be accessed at https://www2.cesm.ucar. edu/models/cesm1.2/cesm/doc/usersguide/x290.html.

Code changes that were made to the source code for this study can be accessed here: https://doi.org/10.7910/DVN/AV1MN3

**Data Availability**

All data used in this study are publicly available. MLS data: https://disc.gsfc.nasa.gov/datasets?page=1&source=Aura%20MLS; ACE-FTS data: http://www.ace.uwaterloo.ca (with registration: https://databace.scisat.ca/l2signup.php); OMPS data: https://disc.gsfc.nasa.gov/datasets?page=1&source=Suomi-NPP%20OMPS; CESM1-CARMA: https://doi.org/10.7910/DVN/AV1MN3

**Competing Interests**

The authors declare that they have no conflict of interest

**Author Contributions**

K. S. and S. S. formulated the study. K. S. did the analysis and wrote the manuscript. P. Y., D. K. helped develop the model. K. S., S. S., P. Y., D. M. M., J. G., and D. K. engaged in discussions and editing of the manuscript.

**Acknowledgments**

K.S., SS. And J. G. gratefully acknowledge support from the atmospheric chemistry division of the National Science Foundation under grant 2316980. D.K. is funded in part by NASA (grant no. 80NSSC19K0952).

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
