# Peer review of "Two-years of stratospheric chemistry perturbations from the 2019/2020 Australian wildfire smoke"

_EGUsphere, 2024_

## Author Comment (AC1)

**We would like to thank the reviewers for their careful review of the paper. Responses to reviewer comments are in bold. Response line numbers correspond to the tracked changes document. Reviewer line numbers correspond to the initially submitted paper.**

**General Comments**

The paper presents interesting and new model studies on heterogeneous chlorine activation on wildfire smoke particles. This includes different sensitivity studies on microphysical interaction with background stratospheric organic aerosol. Unfortunately some figures can convey the message that the effects on ozone are minor in the model and somehow not consistent to the observations. Several times important information is missing.

**Thank you for your comments. We have updated our description on the differences in ozone anomalies in the polar region between the model and the observations as well as added in more description on the calculation of the anomalies. Please see more detail below in the responses to specific comments.**

**Specific Comments**

Line 12: Insert 'in 2020 and 2021'

**Thanks, done.**

Line 103: Please mention the selected retrieval method for OMPS-LP (NASA, USask…) here since this can cause large differences (see also line 226).

**Thanks, added in on lines 105-106: "The Ozone Mapping and Profiler Suite (OMPS) retrievals of aerosol extinction at 675 nm from NASA Goddard space flight center."**

Line 108:  Is an existing transient simulation (with or without nudging) used for initialization on 29 December? This is critical for the correct distribution of halocarbons and total inorganic chlorine and bromine. Which halocarbons are included (including lumped ones)?

**Yes, we initialize from a spin up specified dynamics simulation. We have added this description in the paper here:**

**Line 113-114: Added in: "The model is spun up from a specified dynamics simulation from midsummer 2019 until December 29 2019."**

**The model has a good representation of total Cl and total Br as well as comprehensive halocarbons.**

Line 111: Self-lofting can occur also with weak nudging but your option might be the cleaner approach.

**Thanks, we agree.**

Line 124 or earlier: Which emission inventory is used for background organics? Why is there a problem in Southern midlatitudes?

**Thanks for this comment. The emissions of SOA in the model are based of Yu et al., (2019). The overestimation of SOA in the model in the Southern Hemisphere only affects our heterogeneous chemistry significantly when we include it in the solubility linearization, which is the reason why we have a case of SOA/4 to align better with available observations. The background SOA does not have a significant effect on the case presented in the main paper. We have updated Figure S2 as we applied an incorrect conversion to mass density when creating the figure (not in the simulations). This correction does not affect the overestimation of SOA in the lower stratosphere. See below for new Figure.**

[Figure]

**Figure R1. New Figure S2 with conversion correction.**

Line 165: Organics are a mixture of species with different molecular weights. How is mass here defined? Variable in space and time? Or assumptions (e.g. molecular weight of hexanoic acid)? More details please since heterogeneous chemistry depends on molecular weight.

**Organic mass is taken from the model (CARMA). For wildfire organic mass, this is a sum of the total mass from all 20 size bins (which we injected), it is therefore variable in space and time. For background organics (as SOA) which is not separated by size, organic mass is taken from the model directly. Therefore, it is also variable in space and time. If laboratory measurements of heterogeneous chemistry on organics become available, then using molecular weight of the organics will be important. However, here, we are only using the mass to calculate acidity or a sulfate to organic ratio and therefore the organic molecular weight is not used directly in the heterogeneous chemistry calculations. To make this clearer we have added in the following:**

**Line 140-141: Added in: "Secondary Organic Aerosols (SOA) using the volatile organic compound precursors: isoprene, monoterprene, benzene, xylene, and toluene".**

**Line 196-197: Added in : "...plus the mass of all ANY wildfire organics (taken as a sum of mass from all 20 size bins). Note the organic molecular weight is not needed to adjust the $H_2SO_4$ weight percent"**

Line 177: Anomalies here still undefined. Provide selected time reference periods for observations (MLS 2004-2019?) and model (best it should be the same number of years, e.g. 2004-2019 transient, to avoid artifacts which show up in case of a much smaller number of years).

**Thanks for pointing this out. We have added in explanation of anomalies in the following locations:**

**Line 208-210: "as daily anomalies from a control run for the model (where only SOA is considered when assuming homogenized mixed aerosols for HCl solubility) and from the observed daily mean climatology (2004–2019) for MLS."**

**Line 319-323. Added in: "....daily mean anomalies at 68 hPa and 100 hPa over 2020 and 2021 for CARMA-CESM1, MLS, and monthly mean anomalies for ACE-FTS. The anomalies for CESM1-CARMA are the difference from the control while the anomalies for the observations are the difference from MLS or ACE-FTS respective climatologies over 2004–2019. See figure S4 for absolute values."**

**Figure 2 Caption: "MLS and ACE anomalies are difference from their respective**

climatologies (2004–2019). CESM1-CARMA anomalies are the differences from the control.”

**Figure S6 Caption: “**
**Observed anomalies are differences from daily mean climatology for MLS and monthly mean climatology for ACE-FTS. Modelled anomalies are daily differences from control.”**

Line 180: I don't understand this unexpected behavior in 2021. It can be also due to an ill definition of anomaly.

**Thanks, we agree this is a little confusing. The explanation for the behavior is supplied in the remainder of the paragraph. However, we have updated the explanation to make it clearer:**

**Line 215-218: Updated “However, in 2021, as wildfire organic concentrations continue to diminish, the extra mixed sulfate causes the organic to sulfate ratio to be lower than the control run, resulting in a more acidic mixed particle and lower HCl solubility.” to “However, in 2021, as wildfire organic concentrations continue to diminish, the extra mixed sulfate causes the organic to sulfate ratio to be lower in the wildfire run compared to the control run. This results in a more acidic mixed particle in the wildfire case compared to the control case and therefore lower HCl solubility.”**

Line 211: This depends critically on initialization. Don't speculate here.

**We agree and have removed the following partial sentence:**

**Line 254“... is a difference in the absolute values of HCl compared to observations of around -0.16 ppb.”**

Line 228: Isn't it dangerous to compare different time periods for estimation of anomalies of different quantities? This can introduce additional uncertainties.

**We agree that it likely does include additional uncertainties. However, given the variable nature of stratospheric aerosols due to sporadic volcanic eruptions, and the sheer magnitude of the wildfires, we believe our method gives an adequate qualitative comparison to observations. We have included some additional information regarding the volcanic eruptions that occurred over the time-period shown that could have some influence on any differences between the observations and the model anomalies.**

**Line 283-284: Added in “Volcanic eruptions that occurred in 2019, 2020, and 2021 that also increased the extinction in the observations likely include Ulawun in June and**

**August 2019, Taal in January 2020, and La Soufrière in April 2021 (Asher et al., 2024; Yook et al., 2022)"**

**Line 285-289: Added in: "which could be due to enhancement in aerosol extinction from volcanic sources which is removed in the model anomalies as both the control and wildfire runs include volcanoes. For example, the influence of Ulawun can be seen in late 2019 in Figure 1 where OMPS-LP extinction is elevated before the ANY wildfires. The influence of La Soufrière can be seen in July 2021 in Figure 1a, and 1b where OMPS-LP extinction levels start increasing again in contrast to the model"**

Line 274: Why is there no effect on ozone in 2021 in contrast to observations?

**The model anomaly is the difference from control where both runs use specified dynamics. The MLS anomaly is difference from climatology, so dynamical variability is still present. The MLS anomaly shows 2020 and 2021 are unusually large ozone holes and the model anomaly shows that these unusually large holes are likely not due to chemistry, but dynamics. To avoid confusion around ozone and to allow more accurate comparison of HCl we are now showing Figure 4 as absolute values. The anomaly figure is now Figure S6. Please see later explanation for more detail.**

Line 276: Expand caption: What is shown as greyshading? Variability of observations? Timeframe? I suppose from text that ACE is monthly and the other curves daily. Please mention for clarity.

**Thanks, added in to Figure 2,4, S4, and S6 captions: "The grey shading shows the MLS and ACE-FTS variability."**

Line 320 or earlier: Did the model reproduce the self-lofting smoke filled and ozone poor anticyclonic vortices mentioned in the introduction?

**Thanks for pointing this out. Our model does not simulate the vortices. We have included a sentence about this here:**

**Line 117-118: Added in "However, the model does not simulate the anticyclonic vortices that put some aerosol into the middle stratosphere.**

Line 328: 'and absence of sunlight?'

**Thanks, added in on line 416**

Figure 4: Years without ozone hole dominate the variability, that is somewhat distracting. I don't understand why there is no ozone depletion except for the small response in June in contrast to MLS. Dynamics cannot explain that, especially not for the nudged simulations. Please elaborate. Is there some artifact due the used anomaly method? From this figure

you get the impression that heterogeneous chemistry on organics does not matter for Antarctic ozone in contrast to other studies and the conclusions. Maybe an additional conventional time series plot in the supplement with model (mostly nudged) and observations can help here.

**Figure 4 showed the anomalies of the specified dynamics wildfire run with respect to a specified dynamics control. The size of the ozone hole is predominantly controlled by temperature and as both the wildfire and control simulations use specified dynamics the ozone anomaly is minor in the spring. This does not mean that wildfires didn't cause a large ozone hole through radiative dynamical forcing or chemical-radiative feedbacks. These effects are just captured in the specified dynamical setup for both control and wildfire runs. This contrasts with the observations where the anomaly is from climatology. To avoid confusion, we are now showing absolute values in Figure 4 where it is clear that the ozone hole is large in observations and in the both the model control and wildfire cases. We removed ClO as in the new figure 4 because we can't accurately compare absolute values of MLS ClO and model daily average ClO due to the ClO diurnal cycle and MLS overpass times (ClO is still shown in Figure S6). The anomaly Figure is now Figure S6 (Shown as Figure R2 below). We also changed the latitude band to 80-70S to better capture the core of the vortex, same for Figure 1. We have also added in the following further description:**

**Line 435-441 Changed this sentence: "Since the model uses specified dynamics only chemical effects are captured. Therefore, since there is virtually no difference in austral spring ozone in the control run compared to the wildfire run, it is clear that an abnormally stable and persistent vortex during both 2020 and 2021 contributed to the large and prolonged springtime ozone loss, as seen in the MLS observations."**

**to**

**"For the austral spring, since the model uses specified dynamics for both the control and wildfire cases, only direct chemical effects are captured. Thus, analyzing differences in this way cannot distinguish any chemical-radiative feedback, but since there is virtually no difference in austral spring ozone in the control run compared to the wildfire run, it is clear that an abnormally stable and persistent vortex during both 2020 and 2021 contributed to the large and prolonged springtime ozone loss. This is seen in the MLS observations which show large differences from MLS climatology from October to December in 2020 and 2021 but good agreement with the model.**

**Updated Figure S5 (now Figure 4) to include ozone. See Figure R2 below**

**80–70ºS**

[Figure]

**Figure R2. New Figure 4.**

Supplement: Fig S1: What is 'volcanic background'? Is the value for January 1 already perturbed?

**I can't find any reference to volcanic background in Fig S1? I believe the reviewer is referring to Fig S3 and likely Fig 1? If so, the observed elevated extinction before January 1 is due to the limited climatology and the variable background state of stratospheric sulfur due to volcanic eruptions. There were multiple volcanoes that had already slightly increased the Southern Hemisphere aerosol extinction. We have included a short discussion on this in relation to Figure 1 as described earlier:**

**Line 282-283: Added in "Volcanic eruptions that occurred in 2019, 2020, and 2021 that also increased the extinction in the observations include Ulawun in June and August 2019, Taal in January 2020, and La Soufrière in April 2021 (Asher et al., 2024; Yook et al., 2022)"**

**Line 285-289 Added in "which could be due to enhancement in aerosol extinction from volcanic sources which is removed in the model anomalies as both the control and wildfire runs include volcanoes. For example, the influence of Ulawun can be seen in late 2019 in Figure 1 where OMPS-LP extinction is elevated before the ANY wildfires. The influence of La Soufrière can be seen in July 2021 in Figure 1a, and 1b where OMPS-LP extinction levels start increasing again in contrast to the model."**

**Technical Corrections**

Fig S5: The legends for MLS and ACE have the wrong time.

**Thanks, fixed**

**References**
**Asher, E., Baron, A., Yu, P., Todt, M., Smale, P., Liley, B., Querel, R., Sakai, T., Morino, I., Jin, Y., Nagai, T., Uchino, O., Hall, E., Cullis, P., Johnson, B., and Thornberry, T. D.: Balloon Baseline Stratospheric Aerosol Profiles (B$^2$ SAP)—Perturbations in the Southern Hemisphere, 2019–2022, J. Geophys. Res. Atmospheres, 129, e2024JD041581, https://doi.org/10.1029/2024JD041581, 2024.**

**Yook, S., Thompson, D. W. J., and Solomon, S.: Climate Impacts and Potential Drivers of the Unprecedented Antarctic Ozone Holes of 2020 and 2021, Geophys. Res. Lett., 49, https://doi.org/10.1029/2022GL098064, 2022.**

**Yu, P., Davis, S. M., Toon, O. B., Portmann, R. W., Bardeen, C. G., Barnes, J. E., Telg, H., Maloney, C., and Rosenlof, K. H.: Persistent Stratospheric Warming Due to 2019–2020 Australian Wildfire Smoke, Geophys. Res. Lett., 48, https://doi.org/10.1029/2021gl092609, 2021.**

---

## Author Comment (AC2)

**We would like to thank the reviewer for their careful review of the paper. Responses to reviewer comments are in bold. Response line numbers correspond to the tracked changes document. Reviewer line numbers correspond to the initially submitted paper.**

The manuscript be Stone et al. is a follow-up study based on the paper by Solomon et al. (2022) which demonstrates the heterogeneous chemistry on organic aerosols is important for explaining the observations of chlorine compounds in the air contaminated by bushfire exhaust of the Australian New Year (ANY) wildfires in late December 2019 and early January 2020. In the current manuscript three different model setups for the handling of HCl solubility in the aerosols are discussed.

1. homogeneous mixture of background sulfate aerosol and ANY wildfire organic aerosol
2. separate treatment of background sulfate aerosol and ANY wildfire organic aerosol
3. liquid-liquid-phase separation (LLPS) only in the ANY aerosols

It seems that the last setup reproduces the observations best. This simulations involved a liquid-liquid phase separation only in the air influenced by the ANY wildfire exhaust.

The authors show model results and comparison with observations for aerosol extinction, ozone and chlorine compounds HCl, $ClONO_2$, and ClO. The results are only shown as anomalies and not comparison of the absolute model quantities with observations. This at least leaves some suspicion that an absolute comparison of the shown model parameters with the observations does not look well. Therefore this absolute comparison should be shown to (hopefully) gain confidence in the suggested parametrisations.

I recommend recommend this paper for publication after this point has been clarified and also the following issues have been addressed.

**Thank you for your review. We showed absolute values for the polar region in figure S5 (now Figure 4) for the SH polar region. However, we didn't show it for the midlatitudes. This is now included in a new Figure S4. We have also added in a sentence directing the reader to this figure if they would like to look at the absolute values. Shown below for reference is Figure R3.**

**Line 322. Added in: "See figure S4 for absolute values."**

[Figure]

**Figure R3. New Figure S4 showing midlatitude absolute values.**

**Major Issues**

It seems that the third set of assumptions is the best and is discussed in the main part of the paper, while the results from the other assumptions are only shown only in the supplementary material. It was not evident to me how the presented three assumptions with respect to HCl solubility relate to the original simulation in Solomon et al. (2022). Is one of them identical or are they all different? The advantage of a hybrid model setup with 2 months free running and specified dynamics (SD) afterwards is not clear. Was it proven, that in SD run with nudged winds the self-lofting of the smoke plume is not present? On the other hand, can you show, that in the presented simulation, this effect is well simulated? Lestrelin et al. (ACP, 2023) and Selitto et al. (ACP, 2023) showed the dynamics of developing vortices from the ANY fires. This should be mentioned in the introduction and it would also be nice to see, how well the model describes these vortices. Furthermore, it would be nice to see, if the observed plume structures are reproduced by the model.

Thanks for this comment. The case that is most similar to Solomon et al is the case that is presented in the main paper. We have added in a short description of this:

Line 266-267: Added in: "Note that this method is most similar to what was presented in Solomon et al. (2023), but differs through the linearization of organics to background aerosols, as outlined above."

Regarding the self-lofting and SD simulations. We have added in the additional references regarding the anticyclonic vortices in the existing introduction discussion on lines 30-31. Our simulations follow those from Yu et al. (2021) using 2.5% black carbon that showed consistent self-lofting compared to observations. We have expanded the following sentence to make this clearer.

Line 117-118. Changed: "As we begin the simulation in free running mode, the smoke can self-loft due to the inclusion of 2.5% black carbon (Yu et al., 2021)" to "As we begin the simulation in free running mode, the smoke can self-loft due to the inclusion of 2.5% black carbon which was shown by Yu et al. (2021) to compare well with the observed amount of self-lofting".

If the model is run in SD it is not possible to get the appropriate radiative dynamical feedback that occurs intrinsically in a free-running model. Therefore, an SD simulation may be able to replicate some of the self-lofting due to the dynamical temperatures that are baked into the reanalysis, but self-lofting will be much more realistic in free-running mode. Please see Figures R1 and R2 below where we show the differences between a simulation that is fully SD and one that is in free-running mode until March 1 (same as in the paper). It is clear that running in full SD hinders the self-lofting significantly.

[Figure]

**Figure R1. Model aerosol extinction for the 2-month free running case (top), and full SD case (bottom).**

[Figure]

**Figure R2. Model aerosol extinction difference for the 2-month free running case compared to the full SD case.**

**line 94ff and figs 2e,f and 4e,f:** In fact, it is not recommended to use daytime minus nighttime MLS ClO measurements for polar latitudes. ClO has a significant diurnal cycle with maxima near the local noon and typically near zero values during the night. Further, MLS has a coverage that typically observes at very similar local times for a given latitude. Therefore an average of MLS at a certain latitude would give the mean value for the two corresponding characteristic local times, which is different than a diurnal average calculated by the model. Therefore, for a meaningful comparison, one should calculate model output for the given MLS observation locations and times.

Besides the sampling effect discussed above, it is not at all clear, if the very small anomaly number is realistic, given the precision and accuracy of the MLS measurements. To me, it seems meaningless to show this ClO comparison and I would suggest leaving it out.

**Yes, we agree with this comment. We have corrected the statement regarding MLS CLO by including "in the midlatitudes" on Line 95. We also agree using MLS overpass times and locations would give more accurate results. However, since we are primarily investigating anomalies in the midlatitudes, any large biases due to the two MLS local**

time sampling at any point are largely removed. Please see Figure R3 and R4 below for a comparison of a simulation that has MLS overpass output and is therefore representative of MLS locations and times of observation. While there are noticeable differences in the anomalies, they are small in both the midlatitudes and polar regions. Therefore, we still believe that including ClO using the model daily averages is a valid approach and it is important to show that chlorine is being enhanced beyond what is typically seen so we are opting to keep ClO analysis in the paper.

[Figure]

Figure R3. Comparison of Model ClO anomalies using MLS overpass locations and times for 2020 Southern Hemisphere midlatudes.

[Figure]

**Figure R4. Comparison of Model ClO anomalies using MLS overpass locations and times for 2020 Southern Hemisphere polar region.**

**line 226ff** (and elsewhere): What do you mean exactly by "anomaly"? The difference between the values and the long running mean value? Or its difference with the mean annual cycle? It sounds like the latter, but please clarify that.

**Thanks, yes, it is the latter. We have added in:**

**Line 265: "daily mean anomalies (difference of each day from daily climatologies)".**

Minor issues
**l. 93:** Please explain what you mean by PressureZM

**Added in on line 96: "(zonal mean values on pressure levels)"**

**l. 94:** It is not true that the use of daytime minus nighttime ClO is recommended for observations in high polar latitudes

**Yes, you are correct. We have changed the statement**

**Line 95: Added in "in the midlatitudes"**

**l. 98:** Please use ACE-FTS instead of ACE here and throughout the paper as the ACE satellite does carry other experiments as well

**Thanks, we have changed ACE to ACE-FTS throughout the manuscript and in the figure legends and captions.**

**l. 101:** explain OMPS-LP

**Thanks, we have explained the acronym and the source of retrieval.**

**Line 105-106: "The Ozone Mapping and Profiler Suite (OMPS) retrievals of aerosol extinction at 675 nm from NASA Goddard space flight center"**

**l. 116:** What are primary organic aerosols in primary organic sections? Please explain better, such that a reader may understand the very basic principle without having to read the aerosol model description papers.

**Good point. We have added in the description:**

**Line 133-134: "(such as organics emitted directly into the atmosphere through the wildfires) in both mixed aerosols and primary organic sections. The primary organic section only contains primary organics, while the mixed section contains a mixture of sulfate and organics, as well as salt and dust".**

**l. 120:** What do you mean by $1e^{-6}$ ? Likely $10^{-6}$ , or really $e^{-6}$ ?

**Thanks. To avoid confusion here, we have changed to $10^{-6}$**

**l. 128-131, 305-307:** Please use proper arrows in chemical reactions.

**Thank you. Fixed all equations.**

**l. 134:** change to "HOBr"

**Thanks, fixed**

**References**

**Solomon, S., Stone, K., Yu, P., Murphy, D. M., Kinnison, D., Ravishankara, A. R., and Wang, P.: Chlorine activation and enhanced ozone depletion induced by wildfire aerosol, Nature, 615, 259–264, https://doi.org/10.1038/s41586-022-05683-0, 2023.**

**Yu, P., Davis, S. M., Toon, O. B., Portmann, R. W., Bardeen, C. G., Barnes, J. E., Telg, H., Maloney, C., and Rosenlof, K. H.: Persistent Stratospheric Warming Due to 2019–2020**

Australian Wildfire Smoke, Geophys. Res. Lett., 48, https://doi.org/10.1029/2021gl092609, 2021.

---

## Editor Decision (ED1)

130 are contained in mixed aerosols for calculation of c

bins are also included.

Heterogeneous chemistry reactions in stratospheri

$ClONO_2 + H_2O \quad HNO_3 + HOCl$

135 $ClONO_2 + HCl \quad _2Cl \; HN_3$

$HOCl + HCl \quad _2OH + Cl_2$

$HOBr + HCl \quad BrCl_2 \oplus H$

Reactions R1-R3 follow Shi et al. (2001), and rea

140 (2003) and Waschewsky and Abbatt, (1999) report

resulting in different values for the second order r

80 70";

[Figure]

HCl, 100 hPa

(b)

ppb

3

HCl, 68 hPa

HCl, 68 hPa

CESM1-CARMA wildfire
CESM1-CARMA control
MLS 2020
MLS 2004-2019

(a)

ppb

3

---

## Author Response (AR2)

**We would like to thank the reviewers for their additional comments which have further improved the paper. Author response to reviewer comments is in bold.**

**Review 1**

General Comments
The paper has improved a lot and is better to understand with the revised figures. There are still some minor issues concerning clarity.

Specific Comments
Line 106: Can be misleading. Do you mean "measurements of aerosol extinction at 675 nm using the retrieval method of the NASA Goddard space flight center" or just refer to the data base?

**Changed to reviewer's suggestion to avoid any confusion**

Move "using MERRA2 reanalysis (Gelaro et al., 2017)" from line 119 to line 113. Is the nudging applied to the whole atmosphere or only the lower part of it?

**We have moved the reference to when specified dynamics is first mentioned as the reviewer suggested. We have also included more information of what the specified dynamics is doing.**

**Line 113: added in "…using MERRA2 reanalysis of winds and temperatures"**

Line 114: A remark on ClY and BrY and included halocarbons should be included here (if the list is too long, in the supplement) to avoid the speculations in line 254. In the AGAGE-database are plenty of halocarbons to use for quantification and boundary condition.

**Added in on line 114: "Emissions of ozone depleting substances are from CMIP5 (Meinshausen et al., 2011). In the southern hemisphere midlatitudes, the model total column inorganic chlorine ($Cl_y$) is between $4\times10^{15}$ and $5\times10^{15}$ molecules/cm$^2$, and inorganic bromine ($Br_y$) is between 19-21 ppt in good agreement with the observed and inferred values (WMO, 2022).**

Line 166: Mass ratio or? Please be precise.

**Changed to mass ratio on line 162.**

Line 197: Why? This does not look like external mixing. In the formulae by Solomon et al (2023) the molecular weight of hexanoic acid is used for organics as in line 245. Also in Shi et al. (2001) molecular weights are needed.

**Shi et al. (2001) uses molecular weight of H2SO4 to convert from H2SO4 molality (mol kg-1) to weight percent, which is defined as the mass of H2SO4/mass of solution\*100. Therefore, when we assume organics are mixing with the sulfate aerosols, we only need the organic mass to adjust the H2SO4 weight percent, we don't need molecular weight.**

**In Solomon et al. (2023), the molecular weight of hexanoic acid was used to convert from organic mass fraction solubility to molarity for conversion to Henry's solubility, which doesn't apply here.**

Line 236: "mass fraction" not clear.

**Line 215: Change from "mass fraction of background organics to the mass fraction of sulfate in mixed aerosols" to "mass fraction of background organics to sulfate in mixed aerosols".**

Line 297 and Figure 1: Is this information not in MERRA2 that you have to use ERA5 here?

**We have changed to MERRA2 instead of ERA5 for Figure 1 and Figure S3 10 hPa 60S zonal wind transition time. The results are identical, but it is now consistent with the reanalysis used in the specified dynamics model runs. It was a good catch by the reviewer.**

Line 432: Isn't there a free running part for January and February 2020 (line 116)? Is that only for the wildfire case? Please be precise here. With 'specified dynamics' effects of radiative heating on dynamics are underestimated or almost suppressed, especially if the nudging includes the whole stratosphere.

**Yes, it is free running during January to February only for the wildfire case, However, the statement is discussing austral spring, a time period when specified dynamics is used for all simulations.**

**The wildfire smoke radiative heating is not captured. However, the specified dynamics will have the radiative heating feedbacks baked in on the large scale. We feel that no change needs to be made here.**

Supplement, Figure S1, caption: Mention that sulfate includes 3 volcanic eruptions. The figure is referenced before this is explained in the text (one page later).

**Thanks, added in: Note that both control and wildfire simulations also have emissions from three volcanic eruptions (see main text).**

Technical Corrections
Line 54: Replace "very lowmost" by "lowermost".

**Thanks, fixed.**

Lines 152 to 155: Typos in citations, remove '.'.

**Removed the commas, thanks.**

References: Please use subscripts for chemical species. In Bernath et al. (2022) journal, issue and link are missing.

**Thanks, fixed.**

Supplement: Figure S4 (new): pressures in caption inconsistent with panel labels and titles.

**Thanks, fixed.**

Author Response: Title of Figure R4 (reviewer 2) contains wrong latitude.

**My apologies.**

**Reviewer 2**

The revised manuscript did take into account all of my suggestions. I think it is much better to clearly point out that the model is able to absolutely reproduce the observations. I am still somewhat puzzled by the
very low numbers of ClO anomaly much below the MLS accuracy, that also seem to be reproduced, so I am happy to leave them in the paper.

I would ask you to add some more labels in the figures:
- in fig 4e/f it is not clear which ozone data you are showing, likley MLS, similarly in fig S4

**The reviewer makes a good point here. We have added in more legends to make it clear what observations we are showing each subplot pair.**

In my printout there are some strange misalignments and formatting issues. They will likely be solved when type-setting, just in case I send screenshots from page 4 and 14 to the editor, as I cannot attach them here

**Thank you! I have looked at the attached misalignments and I am not sure what caused them. They look fine to me in both word and pdf format, so hopefully should be fine during type-setting.**